# Examination of the independent contribution of rheumatic heart disease and congestive cardiac failure to the development and outcome of melioidosis in Far North Queensland, tropical Australia

**Phoebe Davies**[1], **Simon Smith**[1], **Rob Wilcox**[2], **James D. Stewart**[1], **Tania J. Davis**[1], **Kylie McKenna**[2], **Josh Hanson**[1,3]*

1 Department of Medicine, Cairns Hospital, Cairns, Queensland, Australia, 2 Tropical Public Health Service, Cairns, Queensland, Australia, 3 Kirby Institute, University of New South Wales, Sydney, Australia

* jhanson@kirby.unsw.edu.au

**Data Availability Statement:** Data cannot be shared publicly because of the Queensland Public

## Abstract

### Background

Patients with rheumatic heart disease (RHD) and congestive cardiac failure (CCF) are believed to have an increased risk of melioidosis and are thought to be more likely to die from the infection. This study was performed to confirm these findings in a region with a high incidence of all three conditions.

### Principal findings

Between January 1998 and December 2021 there were 392 cases of melioidosis in Far North Queensland, tropical Australia; 200/392 (51.0%) identified as an Indigenous Australian, and 337/392 (86.0%) had a confirmed predisposing comorbidity that increased risk for the infection. Overall, 46/392 (11.7%) died before hospital discharge; the case fatality rate declining during the study period (p for trend = 0.001).

There were only 3/392 (0.8%) with confirmed RHD, all of whom had at least one other risk factor for melioidosis; all 3 survived to hospital discharge. Among the 200 Indigenous Australians in the cohort, 2 had confirmed RHD; not statistically greater than the prevalence of RHD in the local general Indigenous population (1.0% versus 1.2%, p = 1.0). RHD was present in only 1/193 (0.5%) cases of melioidosis diagnosed after October 2016, a period which coincided with prospective data collection. There were 26/392 (6.6%) with confirmed CCF, but all 26 had another traditional risk factor for melioidosis. Patients with CCF were more likely to also have chronic lung disease (OR (95% CI: 4.46 (1.93–10.31), p<0.001) and chronic kidney disease (odds ratio (OR) (95% confidence interval (CI): 2.98 (1.22–7.29), p = 0.01) than those who did not have CCF. Two patients with melioidosis and CCF died before hospital discharge; both were elderly (aged 81 and 91 years) and had significant comorbidity.

Health Act 2005. Data are available from the Far North Queensland Human Research Ethics Committee (contact via email Cairns_Ethics@health.qld.gov.au) for researchers who meet the criteria for access to confidential data.

**Funding:** The authors received no specific funding for this work.

**Competing interests:** The authors have declared that no competing interests exist.

## Conclusions

In this region of tropical Australia RHD and CCF do not appear to be independent risk factors for melioidosis and have limited prognostic utility.

## Author summary

Melioidosis, a disease caused by *Burkholderia pseudomallei*, rarely develops in the absence of well-described predisposing conditions that include diabetes mellitus, hazardous alcohol intake, chronic kidney disease, chronic lung disease, malignancy, and immunosuppression. These comorbidities are also strongly linked to patients' short and long-term outcomes. In the large Darwin Prospective Melioidosis Study (DPMS) performed in Australia's Northern Territory, the presence of rheumatic heart disease (RHD) and/or congestive cardiac failure (CCF) were independently associated with pulmonary melioidosis and independently predicted death. Indeed, patients with RHD and/or CCF and melioidosis had the highest case-fatality rate in the DPMS cohort.

The prevalence of RHD and/or CCF in cases of melioidosis in this study in Far North Queensland (FNQ), was similar to that seen in the Northern Territory. However, every patient had at least one other traditional risk factor for the disease. Furthermore, pulmonary involvement and mortality were not higher in patients with RHD and/or CCF. In FNQ, RHD and CCF are not independent risk factors for melioidosis and have limited prognostic utility. The high prevalence of these cardiac diseases in patients with melioidosis may be, at least partly, explained by the confounding presence of socioeconomic disadvantage that increases the incidence of all three conditions.

## Introduction

Melioidosis, an opportunistic infection caused by the environmental, Gram-negative bacterium *Burkholderia pseudomallei*, is endemic in tropical regions, particularly South-East Asia and Northern Australia. The disease usually occurs in people with specific predisposing conditions that include diabetes mellitus, hazardous alcohol intake, chronic kidney disease, chronic lung disease, underlying malignancy, and immunosuppression [1]. Identifying these comorbidities is essential as they are closely linked to long term outcomes. While the case-fatality rate from melioidosis is now <10% in Australia, the five-year mortality of patients with predisposing conditions who survive their melioidosis is up to 23%, higher than that of many cancers [2,3]. Identifying and optimising the subsequent management of these comorbidities is likely to improve patients' long-term outcomes. There are also data to suggest that targeted chemoprophylaxis in populations predisposed to melioidosis might also reduce the incidence of *B. pseudomallei* infection [4].

The Darwin Prospective Melioidosis Study (DPMS)–which commenced in the Northern Territory (NT) of Australia in 1989 –reported that rheumatic heart disease (RHD) and/or congestive cardiac failure (CCF) was almost as common a predisposing factor for melioidosis as immunosuppression (102/1148 (9%) versus 106/1148 (9%)) [5]. In multivariate analysis, the presence of RHD and/or CCF was independently associated with pneumonia and independently predicted death. Indeed, patients with RHD and/or CCF had a higher case fatality rate (19/102, 19%) than patients with any other predisposing comorbidity in the DPMS cohort.

This association between melioidosis and RHD and CCF described in the NT has enormous potential significance in Far North Queensland (FNQ), another region in tropical Australia where the incidence of both melioidosis and RHD are increasing [6,7]. The annual incidence of melioidosis more than doubled between 1998 and 2019 to 9.9/100000 population [8], the annual incidence of RHD has risen to 49/100000 population [6], while the overall prevalence of CCF in FNQ—estimated at 1.5–2%—is one of the highest in Australia [9]. This study was performed to identify the prevalence of RHD and/or CCF among patients diagnosed with melioidosis in FNQ. If a significant association were identified, it would reinforce for local clinicians the importance of seeking the presence of RHD and CCF and optimising their management to improve surviving patients' short and long-term outcomes. It might also influence local public health strategies to prevent the disease.

## Methods

### Ethics statement

The Far North Queensland Human Research Ethics Committee provided ethical approval for the study (HREC/18/QCH/91–1261 and HREC/15/QCH/46–977). As the data were de-identified, the Committee waived the requirement for informed consent.

This study was performed in the FNQ region of tropical Australia which covers an area of over 380,000 km$^2$ and has a population of about 280,000 people, approximately 17% of whom identify as Indigenous Australians [10]. The majority live in the region's urban administrative hub—the city of Cairns—but approximately 10% live in the remote Cape York Peninsula and Torres Strait Islands, a region which borders Papua New Guinea (Fig 1).

All patients in FNQ with culture-confirmed *B. pseudomallei* infection between January 1$^{st}$ 1998, and December 31$^{st}$ 2021, were included in the study. From October 1$^{st}$ 2016, data have been collected prospectively. The patients' medical records were reviewed, and their demographics, medical history and clinical course were recorded. The presence of comorbidities that predispose to melioidosis were specifically sought; these included diabetes mellitus, hazardous alcohol use, chronic kidney disease, chronic lung disease, malignancy, and immunosuppression as previously defined [11]. A history of RHD and/or CCF was also sought. RHD was said to be present if the diagnosis was documented in the medical record or was identified on an echocardiogram by a specialist physician. As acute rheumatic fever (ARF) and RHD are notifiable diseases in Queensland (and have been from 1999 and 2018, respectively), the state's RHD register was also reviewed to ensure that no cases of RHD were missed. CCF was said to be present if the diagnosis was recorded in the medical record, or if systolic or clinically significant diastolic impairment of left ventricular function was identified on an echocardiogram reported by a specialist physician and the patient was receiving diuretic therapy without another obvious indication. Pulmonary melioidosis was defined as the presence of consolidation on chest imaging performed during the patient's episode of melioidosis and/or *B. pseudomallei* cultured from sputum. All individuals receiving care in Queensland's public health system are asked whether they identify as an Indigenous Australian (an Aboriginal Australian, a Torres Strait Islander Australian or both). Australian Bureau of Statistics data were used to determine disease incidence and prevalence [10].

### Statistical analysis

Data were de-identified, entered into an electronic database (Microsoft Excel 2016, Microsoft, Redmond, WA, USA) and analysed using statistical software (Stata version 14.2, StataCorp LLC, College Station, TX, USA). Groups were analysed using logistic regression, the chi-squared, Fisher's exact or Kruskal-Wallis tests, where appropriate. Trends over time were

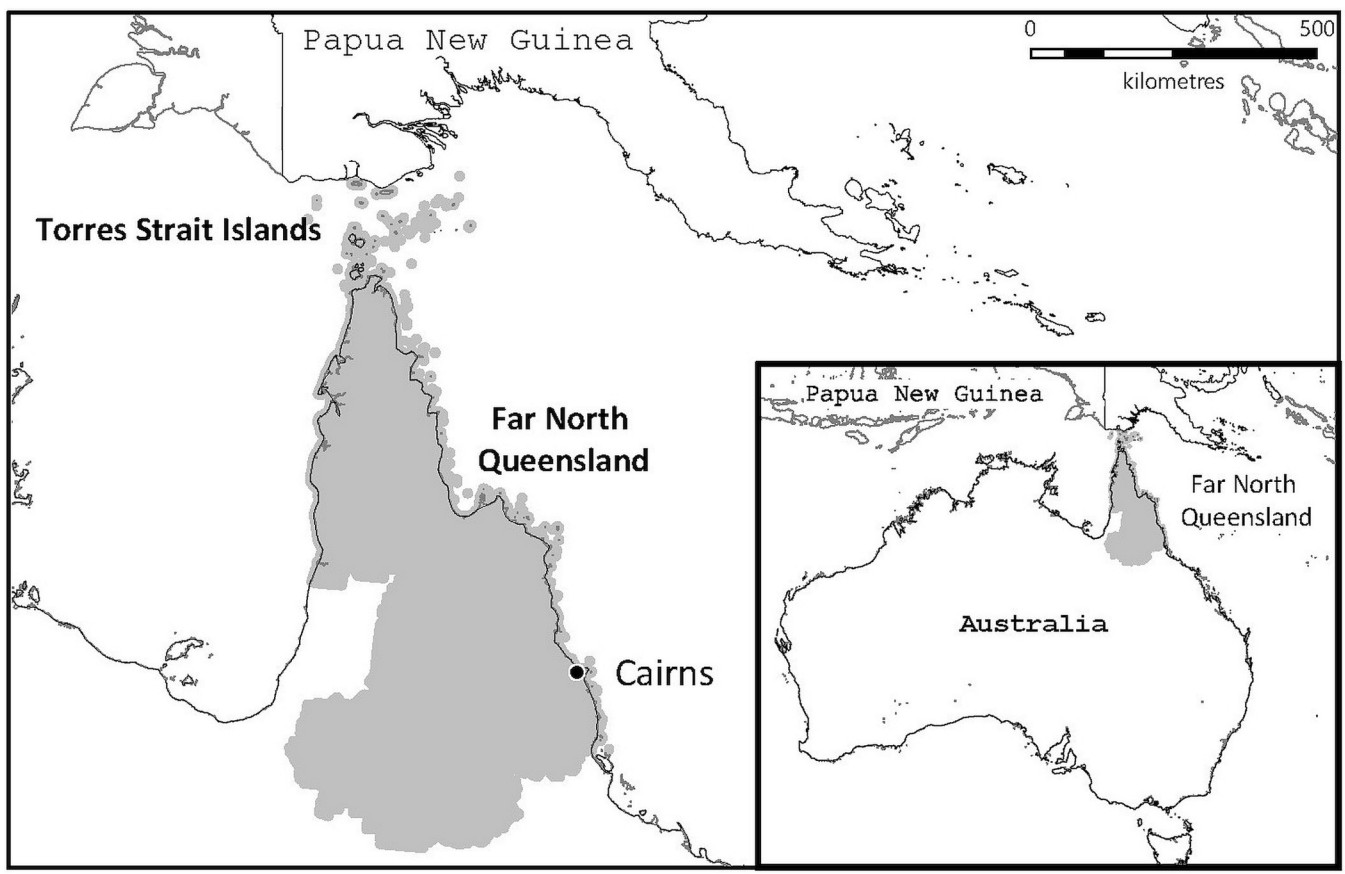

**Fig 1. The study region of Far North Queensland in tropical Australia.** The map was constructed using mapping software (MapInfo version 15.02, Connecticut, USA) using data provided by the State of Queensland (QSpatial). Queensland Place Names—State of Queensland (Department of Natural Resources, Mines and Energy) 2019, available under Creative Commons Attribution 4.0 International licence https://creativecommons.org/licenses/by/4.0/. 'Coastline and state border–Queensland—State of Queensland (Department of Natural Resources, Mines and Energy) 2019, available under Creative Commons Attribution 4.0 International licence https://creativecommons.org/licenses/by/4.0/.

determined using an extension of the Wilcoxon rank-sum test [12]. If individuals were missing data, they were not included in analyses which evaluated those variables.

## Results

There was a total of 392 cases of melioidosis in FNQ during the study period; the mean annual incidence was 6.1/100000 population for the entire FNQ region: 3.8/100000 population in the urban Cairns region and 24.7/100000 population in the remote Cape York/Torres Strait Islands region. The incidence increased over the course of the study and in the final 12 months of the study period, the incidence in FNQ was 17.4/100000/year; 16.2/100000/year in the urban Cairns region and 28.5/100000/year in the remote Cape York/Torres Strait Islands region.

Of the 392 cases, 200 (51.0%) identified as an Indigenous Australian; 337/392 (86.0%) had an identifiable risk factor for melioidosis with diabetes mellitus and hazardous alcohol the most common (Table 1). Overall, 46/392 (11.7%) died before hospital discharge; the case fatality rate declined over the course of the study (p for trend = 0.001). In the final 12 months of the study period 3/50 (6%) of the patients died from their melioidosis (Table 2).

**Table 1. Proportion of the FNQ cohort with the different predisposing risk factors for melioidosis, stratified by Indigenous status.**

|  | All n = 392 [a] | Indigenous n = 200 | Non-Indigenous n = 192 | p |
|---|---|---|---|---|
| Diabetes | 201/377 (53.3%) | 134/188 (71.3%) | 67/189 (35.5%) | <0.001 |
| Hazardous alcohol use | 148/360 (41.1%) | 77/176 (43.8%) | 71/184 (38.6%) | 0.32 |
| Chronic lung disease | 69/363 (19.0%) | 15/178 (8.4%) | 54/185 (29.2%) | <0.001 |
| Chronic kidney disease | 55/375 (14.5%) | 39/186 (20.9%) | 16/189 (8.5%) | 0.001 |
| Rheumatic heart disease [b] | 3/359 (0.8%) | 2/175 (1.1%) | 1/184 (0.5%) | 0.62 |
| Congestive cardiac failure [c] | 26/353 (7.3%) | 10/169 (5.9%) | 16/184 (8.7%) | 0.32 |
| Active malignancy | 35/362 (9.7%) | 8/177 (4.5%) | 27/185 (14.6%) | 0.001 |
| Immunosuppression | 52/243 (21.4%) | 18/103 (17.5%) | 34/140 (24.3%) | 0.21 |
| No risk factor documented [d] | 55/392 (14.0%) | 26 (13.0%) | 29 (15.1%) | 0.56 |

[a] The denominator for each of the risk factors varies due to incomplete data for some of the 197 patients presenting before 2017 in whom data were collected retrospectively

[b] None of the remaining 33 patients were on the Queensland ARF/RHD register

[c] Excluding cases of congestive cardiac failure that were a complication of rheumatic heart disease

[d] Inability to access some of the medical records of the 197 patients presenting before 2017 means that this is likely to be an artificially inflated figure.

There were only 3/392 (0.8%) patients in the cohort with confirmed RHD. There were 205/392 (52.3%) who had an echo report available; the remaining 187 were not on Queensland ARF/RHD register. The medical records of 154 (82.3%) of these 187 were accessible and RHD or RHD therapy were not documented in any. All 3 RHD cases had at least one other risk factor for melioidosis (S1 and S2 Tables, Fig 2). Two of the 3 patients with RHD and melioidosis had severe mitral valve disease, however, the third had only mitral valve thickening with no evidence of stenosis or regurgitation. All 3 patients with RHD survived to hospital discharge.

It was possible to reliably determine the presence of CCF in 353/392 (90.1%); 205 had an echocardiogram and in 148 (79.1%) of the remaining 187 there were sufficient clinical data in the medical record to determine its presence. CCF was confirmed in 26/353 (7.4%) patients,

**Table 2. Comparison of the proportion of melioidosis cases with the different predisposing risk factors in the Darwin Prospective Melioidosis Series (DPMS) and Far North Queensland cohort.**

|  | DPMS n = 1148 [a] | | Far North Queensland n = 392 [b] | |
|---|---|---|---|---|
|  | Number (%) | Died (%) | Number (%) | Died (%) |
| Diabetes | 513 (45%) | 62 (12%) | 201/377 (53%) | 18 (9%) |
| Hazardous alcohol use | 455 (40%) | 56 (12%) | 148/360 (41%) | 15 (10%) |
| Chronic lung disease | 312 (27%) | 45 (14%) | 69/363 (19%) | 9 (13%) |
| Chronic kidney disease | 140 (12%) | 24 (17%) | 55/375 (15%) | 8 (15%) |
| Rheumatic heart disease or congestive cardiac failure | 102 (9%) | 19 (19%) | 29/346 (8%) | 2 (7%) |
| Malignancy | 111 (10%) | 20 (18%) | 35/362 (10%) | 6 (17%) |
| Immunosuppression | 106 (9%) | 18 (17%) | 52/243 (21%) | 5 (10%) |
| No risk factors | 186 (16%) | 3 (2%) | 55/392 (14%) | 14 (26%) [c] |
| Overall | 1148 | 133 (12%) | 392 | 47 (12%) |

[a] DPMS: Darwin prospective melioidosis study. Data collected prospectively from October 1989 [5].

[b] Data were collected prospectively in FNQ after October 2016.

[c] Inability to access some of the medical records of the 197 patients presenting before 2017 means that this is likely to be an artificially inflated figure. It also means that the patients from Far North Queensland with individual predisposing factors recorded were more likely to come from later in the study period at a time when treatment algorithms had been established and ICU care had evolved. This is likely to explain why Far North Queensland patients with individual predisposing risk factors documented appear to have a lower case-fatality rate than the DPMS patients with the same risk factor.

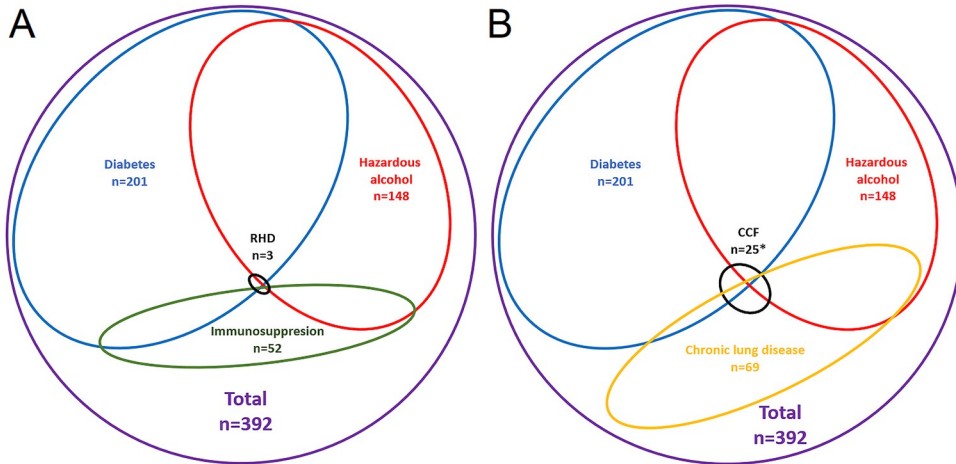

**Fig 2. A Venn diagram showing that all the patients in the cohort with rheumatic heart disease or congestive cardiac failure had at least one additional traditional predisposing condition for melioidosis.** *There was 1 additional patient with CCF who had an active malignancy, but no diabetes, chronic lung disease or history of hazardous alcohol use. RHD: Rheumatic heart disease. CCF: Congestive cardiac failure. Only the patients with diabetes mellitus, hazardous alcohol use, chronic lung disease, immunosuppression, RHD and CCF are shown in the figure. There were, in addition, 53 patients with chronic kidney disease, 35 patients with an active malignancy and 56 patients in whom no risk factor was identified.

21 (80.1%) had left ventricular systolic impairment while 5 (19.9%) had isolated diastolic dysfunction. However, again, all 26 had another risk factor for melioidosis with 13/26 (50.0%) having two or more additional risk factors. Patients with CCF were more likely to also have chronic lung disease (OR (95% CI: 4.46 (1.93–10.31), p<0.001) and chronic kidney disease (odds ratio (OR) (95% confidence interval (CI): 2.98 (1.22–7.29), p = 0.01) than those who did not have CCF (S1 and S3 Tables, Fig 2). There were 2/26 (7.7%) patients with CCF who died before hospital discharge, an 81-year-old non-Indigenous man with ischaemic cardiomyopathy, diabetes mellitus, chronic kidney disease and prostate cancer and a 91-year-old non-Indigenous man with ischaemic heart disease associated with diastolic left ventricular dysfunction, who also had cerebrovascular disease and chronic kidney disease.

Pulmonary melioidosis was present in 23/29 (79.3%) with RHD or CCF, compared to 227/317 (71.6%) who had neither diagnosis (p = 0.52); pulmonary melioidosis was present in 2/3 (66.7%) patients with RHD.

Patients with melioidosis and RHD were younger (aged 37, 39 and 44 years) than those patients with melioidosis and CCF (median (IQR): 68 (51–81) years (p<0.001); 2/3 (66.7%) patients living with RHD were Indigenous Australians versus 10/26 (38.5%) of those with CCF (p = 0.32). Among the 200 Indigenous Australians in this cohort, there were 2 cases of RHD, a prevalence that was not statistically greater than the prevalence of RHD in the general FNQ Indigenous population (1.0% versus 1.2%, p = 1.0) [6]. RHD was present in only 1/193 (0.5%) cases of melioidosis diagnosed since October 2016, a period which coincides with prospective, comprehensive data collection and an increased incidence of melioidosis in the urban Cairns region. Among the patients who had data collected prospectively, 107/193 (55.4%) had an echocardiogram performed and 82/193 (42.5%) identified as Indigenous Australians.

## Discussion

There are many similarities between the patients diagnosed with melioidosis in FNQ and in the Northern Territory of Australia. Indigenous Australians bear a disproportionate burden of

disease in both locations and diabetes mellitus and hazardous alcohol consumption are the most common predisposing factors. However, in this cohort of 392 FNQ patients, it is notable that every patient with melioidosis and confirmed RHD or CCF had another traditional predisposing risk factor for *B. pseudomallei* infection. This is despite the rising local incidence of melioidosis, a significant and growing burden of RHD and a prevalence of CCF that is amongst the highest in Australia.

The proposed pathophysiological mechanism for RHD and/or CCF acting as a predisposing factor for melioidosis is alveolar flooding leading to disruption of the alveolar lining, inhibiting local macrophage and neutrophil function [13]. There are data to suggest that CCF increases the risk of pneumonia and mortality [14–16]. However, while RHD and CCF were found to be associated with pulmonary melioidosis in the NT, this association was not apparent in FNQ. Furthermore, while the presence of either RHD or CCF was independently associated with death from melioidosis in the NT, the only two patients in FNQ with melioidosis and either RHD or CCF to die before discharge in FNQ in over 20 years were both elderly with multiple comorbidities. These data suggest that, in FNQ at least, RHD and CCF have a limited independent role in the pathogenesis of melioidosis and that their presence does not necessarily herald a poor outcome.

The obvious question is why RHD and CCF appear to have a less important pathophysiological role in melioidosis in FNQ than in the NT, considering the many other epidemiological similarities between the two cohorts. Indeed, although the overall prevalence of RHD and/or CCF among patients with melioidosis was similar in both locations, it is striking how infrequently RHD was seen in the FNQ cohort, with only 3 confirmed cases seen in 24 years, 2 of whom had diabetes, the archetypal risk factor for melioidosis, with the third taking two immunosuppressive medications [17].

One potential explanation could be the higher burden of RHD in the NT, which has a prevalence among Aboriginal Australians—who represent 30.3% of the NT population—of 26/1000 [18]. This compares with a RHD prevalence in FNQ among Aboriginal and Torres Strait Islander Australians—who represent approximately 17% of the local population—of 12/1000 [6]. However, this would not explain why the prevalence of RHD among Indigenous FNQ patients with melioidosis was almost the same as the prevalence of RHD in the general Indigenous FNQ population.

Perhaps differences in RHD severity might explain the disparity, as patients with severe RHD would be at a higher risk of developing pulmonary congestion. However, the proportion of cases with severe RHD is similar in the two jurisdictions: 32.7% of the diagnosed RHD in Queensland is severe and 26.9% is moderate, while in the NT 24.7% and 25.6% of RHD cases are severe and moderate respectively [18]. A similar proportion of a greater number of RHD cases in the NT would mean a greater absolute number of patients with severe RHD in the NT, but this would still not seem to explain the infrequency of RHD in patients with melioidosis in FNQ.

Another possibility is that there is heterogeneity in the incidence of RHD and melioidosis across the FNQ region. The recent increase in incidence of melioidosis has occurred predominantly in the urban Cairns region, an area where only 11.6% of the population is Indigenous and where the prevalence of RHD is, accordingly, lower [8]. However, this would not explain the absence of cases of RHD in patients with melioidosis from remote FNQ (Cape York and the Torres Strait Islands). In this region the mean annual incidence of melioidosis over the study period was 24.7/100000 (similar to the mean annual incidence in the DPMS of 20.5/100,000 over the course of that study) and the prevalence of RHD in some communities is as high as 27/1000, similar to that which is reported in Aboriginal Australians in the NT.

The proposed pathophysiological mechanism for the association between RHD and/or CCF and melioidosis is biologically plausible, but an underlying factor which predisposes to both melioidosis and RHD might also contribute to the association between the diseases

described in the NT. Socioeconomic disadvantage, with its impact on the social determinants of health, is linked strongly to the incidence and outcomes of both melioidosis and RHD in FNQ [6,19]. The association between disadvantage and melioidosis is explained by the higher rates of predisposing factors, particularly diabetes mellitus and chronic kidney disease, that is seen in disadvantaged populations [20]. Socioeconomic disadvantage is also more common in rural and remote FNQ where environmental exposure to *B. pseudomallei* may be greater. Although the association between socioeconomic disadvantage and melioidosis has not been studied in detail in the NT, it almost certainly contributes to the higher burden of melioidosis in Indigenous Australians in that jurisdiction [5]. RHD, meanwhile, is one of the classic diseases of disadvantage, where the social determinants of health, particularly household crowding, play a critical role in the pathogenesis of the condition [21].

It could also be hypothesised that socioeconomic disadvantage might contribute to the finding of an association between RHD and mortality seen in the DPMS. When socioeconomic disadvantage was included in a multivariate analysis of prognostic factors in a FNQ study—which included age and Indigenous status—residence in the most disadvantaged areas of the region was the only independent predictor of death in patients with melioidosis, apart from ICU admission [19]. Meanwhile residents of the most disadvantaged regions of FNQ have the greatest burden of RHD but are less likely to receive valve surgery [6], highlighting the paradox that the socioeconomic circumstances that predispose patients to disease, also lead to less access to the sophisticated, multidisciplinary health services that they require [22,23].

Similar trends are seen among patients with CCF and melioidosis in FNQ. All patients with CCF in the cohort had at least one additional traditional risk factor for disease, with chronic kidney disease and chronic lung disease particularly common. This may not be surprising as many of the factors that predispose patients to melioidosis–particularly diabetes, chronic kidney disease and hazardous alcohol consumption–also increase the risk of cardiac disease [24,25]. Pulmonary melioidosis was no more common in the patients with CCF and did not herald a worse outcome. The prevalence of CCF also demonstrates a strong inverse association with socioeconomic disadvantage in Australia [26], suggesting that this may also contribute to the finding of an association between CCF and melioidosis described in the DPMS.

In other parts of the world where melioidosis is endemic and there is a high prevalence of RHD and CCF, studies also suggest they may also have a limited role in the pathogenesis of melioidosis [27]. The incidence of melioidosis in northeast Thailand has been reported to be as high as 21.3/100000 population and there is also a significant burden of CCF and RHD [28,29]. It was notable that neither cardiac condition is reported in a detailed series of 204 cases of *B. pseudomallei* bacteremia or in another review of over 7000 cases [27]. The situation is similar in reports from India and Malaysia [30–35]. In all three countries more traditional comorbidities, particularly diabetes and chronic kidney disease, were the most commonly reported risk factors [32–35].

Our study has significant limitations, the most important of these being retrospective data collection before October 2016 precluding comprehensive data collection prior to this point and potentially missing cases where RHD and CCF were the only predisposing factor for melioidosis. However, the Queensland RHD register was used to cross-reference RHD cases, and it is notable that there was only one case of RHD in the 193 patients diagnosed after prospective, comprehensive data collection commenced. The criteria employed to define CCF were inclusive, with all patients receiving diuretics and having any systolic or diastolic left ventricular dysfunction on echocardiogram receiving this label, however this would tend to overestimate, rather than underestimate, the prevalence of CCF in the cohort. This FNQ cohort is also less than half the size of the DPMS, increasing the likelihood of type 2 errors, however even acknowledging this, the number of RHD cases in a period of over two decades was very low. It was notable that in the FNQ cohort the demographic characteristics of the RHD and

CCF populations were quite different: 2 of the 3 RHD patients were Indigenous while most patients with CCF were non-Indigenous and the median age of the RHD patients was 25 years lower than that of the CCF patients. This may confound analyses which combine these populations; however, we presented the populations together to facilitate comparison with the DPMS. In the DPMS, patients with RHD and CCF are presented together, presumably as both diseases are associated with pulmonary congestion, and it was notable that the prevalence of RHD and/ or CCF among melioidosis cases was similar in the two cohorts. It would be interesting to see if the relative proportions of patients with RHD and CCF in the DPMS and the FNQ cohort were also similar. There were relatively few children in the FNQ cohort and it was notable that while over 80% of the children in a Darwin paediatric series had no identifiable risk factor for melioidosis, one of three deaths occurred in a child with severe RHD, while another death occurred in a child with congenital heart disease, neither had another predisposing comorbidity [36].

As the case-fatality rate from melioidosis declines in Australia, there is a growing focus on the diagnosis of the comorbidities that predispose patients to the infection, which are also associated with subsequent premature, and frequently preventable, death. Indeed, identification of these conditions and optimisation of their subsequent management is now part of routine clinical care of patients diagnosed with melioidosis in Australia [2,3]. Renal impairment will be biochemically apparent, but patients should be screened for diabetes and asked specifically about hazardous alcohol consumption, and symptoms consistent with chronic lung disease and immunosuppression. Age-appropriate cancer screening should also be considered. However, while it would be prudent to ask on systems review about symptoms consistent with CCF and RHD, in FNQ routine echocardiogram appears to be a low-value intervention in the absence of clinical suspicion for cardiac disease.

While the data from this study do not exclude a causal association between pulmonary congestion and melioidosis, socioeconomic disadvantage increases an individual's risk of developing melioidosis and both CCF and RHD and may explain the frequency with which these cardiac conditions are seen in patients with melioidosis, even before pathophysiological mechanisms are proposed. Certainly, a greater focus on the societal inequities that increase the incidence of melioidosis, RHD and CCF is likely to not only reduce their incidence, but also the burden of many other communicable and non-communicable diseases that are disproportionately borne by the most disadvantaged members of our society.

## Supporting information

**S1 Table. Additional risk factors for melioidosis in patients with rheumatic heart disease and congestive cardiac failure.**
(DOCX)

**S2 Table. Association between the presence of rheumatic heart disease and other predisposing factors.**
(DOCX)

**S3 Table. Association between the presence of congestive cardiac failure and other predisposing factors.**
(DOCX)

## Acknowledgments

The authors would like to acknowledge all the health workers who were involved in the care of the patients. They would also like to acknowledge Mr Peter Horne who assisted with the production of Fig 1.

## Author Contributions

**Conceptualization:** Simon Smith, Josh Hanson.

**Data curation:** Phoebe Davies, Simon Smith, Rob Wilcox, James D. Stewart, Josh Hanson.

**Formal analysis:** Josh Hanson.

**Investigation:** Phoebe Davies, Simon Smith, Josh Hanson.

**Methodology:** Phoebe Davies, Josh Hanson.

**Supervision:** Simon Smith, Kylie McKenna, Josh Hanson.

**Writing – original draft:** Phoebe Davies, Josh Hanson.

**Writing – review & editing:** Simon Smith, Rob Wilcox, James D. Stewart, Tania J. Davis, Kylie McKenna, Josh Hanson.

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
