## [Decision Letter · Decision Letter 0]

22 Jun 2022

Dear Dr Hanson,

Thank you very much for submitting your manuscript "Examination of the independent contribution of rheumatic heart disease and congestive cardiac failure to the development and outcome of melioidosis in Far North Queensland, tropical Australia." for consideration at PLOS Neglected Tropical Diseases. As with all papers reviewed by the journal, your manuscript was reviewed by members of the editorial board and by several independent reviewers. The reviewers appreciated the attention to an important topic. Based on the reviews, we are likely to accept this manuscript for publication, providing that you modify the manuscript according to the review recommendations. 

Sincerely,

Joseph M. Vinetz

Deputy Editor

Joseph Vinetz

Deputy Editor

Reviewer's Responses to Questions

**Key Review Criteria Required for Acceptance?**

**Methods**

-Are the objectives of the study clearly articulated with a clear testable hypothesis stated?

-Is the study design appropriate to address the stated objectives?

-Is the population clearly described and appropriate for the hypothesis being tested?

-Is the sample size sufficient to ensure adequate power to address the hypothesis being tested?

-Were correct statistical analysis used to support conclusions?

-Are there concerns about ethical or regulatory requirements being met?

Reviewer #1: No new analyses required

**Results**

-Does the analysis presented match the analysis plan?

-Are the results clearly and completely presented?

-Are the figures (Tables, Images) of sufficient quality for clarity?

Reviewer #1: Results clearly presented

**Conclusions**

-Are the conclusions supported by the data presented?

-Are the limitations of analysis clearly described?

-Do the authors discuss how these data can be helpful to advance our understanding of the topic under study?

-Is public health relevance addressed?

Reviewer #1: Conclusions and limitations clearly described

**Editorial and Data Presentation Modifications?**

Reviewer #1: A few very minor edits need - for example commas not necessary in lines 135, 144, 155 and 356; insert gap before parentheses in first column of 'Hazardous alcohol use' in Table 2.

**Summary and General Comments**

Reviewer #1: This well written paper looks at whether there was a significant association between melioidosis and rheumatic heart disease and congestive cardiac failure in Far North Queensland. The reasons why such as association was not seen in this cohort as opposed to that in Darwin is discussed.

PLOS authors have the option to publish the peer review history of their article (what does this mean?). If published, this will include your full peer review and any attached files.

Reviewer #1: Yes: David Dance

Figure Files:

Data Requirements:

Reproducibility:

References

---

## [Editor Report · Decision Letter 1]

23 Jun 2022

Dear Dr Hanson,

We are pleased to inform you that your manuscript 'Examination of the independent contribution of rheumatic heart disease and congestive cardiac failure to the development and outcome of melioidosis in Far North Queensland, tropical Australia.' has been provisionally accepted for publication in PLOS Neglected Tropical Diseases.

Best regards,

Joseph M. Vinetz

Deputy Editor

Joseph Vinetz

Deputy Editor

---

## [Editor Report · Acceptance letter]

2 Jul 2022

Dear Dr. Hanson,

We are delighted to inform you that your manuscript, "Examination of the independent contribution of rheumatic heart disease and congestive cardiac failure to the development and outcome of melioidosis in Far North Queensland, tropical Australia.," has been formally accepted for publication in PLOS Neglected Tropical Diseases.

Best regards,

Shaden Kamhawi

co-Editor-in-Chief

Paul Brindley

co-Editor-in-Chief
